# Identification of Polyunsaturated Fatty Acids Synthesis Pathways in the Toxic Dinophyte *Alexandrium minutum* Using ^13^C-Labelling

**DOI:** 10.3390/biom10101428

**Published:** 2020-10-08

**Authors:** Marine Remize, Frédéric Planchon, Ai Ning Loh, Fabienne Le Grand, Christophe Lambert, Antoine Bideau, Adeline Bidault, Rudolph Corvaisier, Aswani Volety, Philippe Soudant

**Affiliations:** 1UMR 6539 LEMAR, CNRS, IRD, Ifremer, University of Brest, F-29280 Plouzané, France; frederic.planchon@univ-brest.fr (F.P.); fabienne.legrand@univ-brest.fr (F.L.G.); christophe.lambert@univ-brest.fr (C.L.); antoine.bideau@univ-brest.fr (A.B.); adeline.bidault@univ-brest.fr (A.B.); rudolph.corvaisier@univ-brest.fr (R.C.); 2Center for Marine Science, Department of Earth and Ocean Sciences, University of North Carolina Wilmington, 5600 Marvin K. Moss Ln., Wilmington, NC 28403, USA; lohan@uncw.edu; 3Department of Biology, Elon University, 50 Campus Drive, Elon, NC 27244, USA; avolety@elon.edu

**Keywords:** PUFA synthesis, 22:6n-3, DHA, PKS pathway, Dinophyte, *Alexandrium minutum*, encystment

## Abstract

The synthetic pathways responsible for the production of the polyunsaturated fatty acids 22:6n-3 and 20:5n-3 were studied in the Dinophyte *Alexandrium minutum*. The purpose of this work was to follow the progressive incorporation of an isotopic label (^13^CO_2_) into 11 fatty acids to better understand the fatty acid synthesis pathways in *A. minutum*. The Dinophyte growth was monitored for 54 h using high-frequency sampling. *A. minutum* presented a growth in two phases. A lag phase was observed during the first 30 h of development and had been associated with the probable temporary encystment of Dinophyte cells. An exponential growth phase was then observed after t_30_. *A. minutum* rapidly incorporated ^13^C into 22:6n-3, which ended up being the most ^13^C-enriched polyunsaturated fatty acid (PUFA) in this experiment, with a higher ^13^C atomic enrichment than 18:4n-3, 18:5n-3, 20:5n-3, and 22:5n-3. Overall, the ^13^C atomic enrichment (AE) was inversely proportional to number of carbons in n-3 PUFA. C_18_ PUFAs, 18:4n-3, and 18:5n-3, were indeed among the least ^13^C-enriched FAs during this experiment. They were assumed to be produced by the n-3 PUFA pathway. However, they could not be further elongated or desaturated to produce n-3 C_20_-C_22_ PUFA, because the AEs of the n-3 C_18_ PUFAs were lower than those of the n-3 C_20_-C_22_ PUFAs. Thus, the especially high atomic enrichment of 22:6n-3 (55.8% and 54.9% in neutral lipids (NLs) and polar lipids (PLs), respectively) led us to hypothesize that this major PUFA was synthesized by an O_2_-independent Polyketide Synthase (PKS) pathway. Another parallel PKS, independent of the one leading to 22:6n-3, was also supposed to produce 20:5n-3. The inverse order of the ^13^C atomic enrichment for n-3 PUFAs was also suspected to be related to the possible β-oxidation of long-chain n-3 PUFAs occurring during *A. minutum* encystment.

## 1. Introduction

Phytoplankton, at the basis of food webs, are the main producers of long-chain polyunsaturated fatty acids (LC-PUFAs), 20:5n-3 and 22:6n-3 [1,2,3]. n-3 PUFAs are essential compounds for marine organisms because they are not always able to synthesize them in sufficient quantities and then have to collect them from their diet [4,5]. 20:5n-3 and 22:6n-3 have beneficial effects on development and growth and can even prevent diseases in humans [6,7,8,9,10]. They are likely able to reduce the impact of cardiovascular diseases or diabetes [7,11,12,13]. Because of their valuable properties, the global demand for these two compounds has increased in recent years, and they are now becoming scarce in relation with marine fish stock reduction, overfishing, and global warming, with fish oils being the main source of n-3 LC-PUFAs [14,15,16]. Despite their economic and ecologic interest, the biological and ecological processes responsible for their synthesis are still poorly known. Understanding how 20:5n-3 and 22:6n-3 are produced at the basis of food webs appears to be of prime interest. This would help in characterizing the expected decrease in n-3 LC-PUFA formation in relation to global changes.

20:5n-3 and 22:6n-3 are produced by two different metabolic pathways [17,18,19]. The fatty acid synthase (FAS) pathway, coupled with the elongase and desaturase steps of the n-3 and n-6 pathways and associated ω3-desaturase route [20], are the most “conventional” ones. With acetyl-CoA as a precursor, in aerobic conditions the FAS pathway is responsible for the synthesis of saturated fatty acids by the progressive addition of two atoms of carbon following the successive actions of four enzymes, ketoacyl-ACP synthase (KS), ketoacyl-ACP reductase (KR), hydroxacyl-ACP dehydrase (DH), and enoyl-ACP reductase (ER). The main products of this synthesis route—namely, palmitic acid (16:0) and stearic acid (18:0)—are then further desaturated and elongated by the elongases and desaturases of the known n-3 and n-6 pathways to form more complex n-3 and n-6 PUFAs. The Polyketide Synthase (PKS) pathway consists of an alternative pathway that can also produce n-3 PUFAs but is O_2_-independent. It has been found in bacteria and heterotroph protists such as Thraustochytrids [21,22] and Dinophytes [23,24,25,26]. The PKS pathway relies on the same four enzymes as the FAS pathway. It is less energy consuming because the metabolites used are simultaneously desaturated and elongated, creating rapidly highly unsaturated long-chain fatty acids with no dehydration and reduction steps. PKS enzymes can have different configurations that define different PKS types. Type I PKS has been identified in Haptophytes [27,28], Dinophytes [25,29], and Thraustochytrids [30], and consists of a large multifunctional enzyme presenting all catalytic domains on the same peptide. Type II, discovered in Haptophytes (especially *Emiliania huxleyi*), Cryptophytes, and later in Dinophytes by Kohli et al. [30], and Type III, found in bacteria and brown algae, among others [31,32], are mono-functional enzymes with one catalytic domain. 

The different pathways described above are not always active or even present in all phytoplankton or micro-zooplankton species, resulting in the different fatty acid compositions found in marine primary producers. Diatoms, which are suspected to preferentially use the “conventional” routes [33], are naturally richer in 20:5n-3, while Dinophytes, suspected to use both the “conventional” and PKS pathways, are richer in 22:6n-3. Some species such as Chlorophytes taxa or Cyanobacteria do not synthesize them or do so only in very small proportions [34].

Dinophytes are one of the main protists of marine and freshwater ecosystems [35]. Some of them are responsible for harmful blooms, called red tides, causing both human illness and the large-scale mortality of diverse marine organisms [36]. To control or at least limit their impact on ecosystems and aquaculture, numerous studies have been undertaken to better understand the mechanisms responsible for their proliferation as well as formation, the persistence of toxins, and the role of cysts [37,38,39,40,41,42]. Encystment is part of Dinophytes’ life cycle and has also been developed to cope with changes in their environment [43]. Cysts can be of two forms: temporary cysts are produced in response to short-term stress or environment perturbations, while resting cysts are part of the life cycle of Dinophytes. Temporary cysts can play an important role in maintaining blooms, as they allow the survival of Dinophytes when growth conditions are less favorable [42]. In other species, the germination of resting cysts can be involved in toxic bloom initiation [44]. The PKS pathway was reported to be involved in toxins biosynthesis [28,45,46] and some fatty acids (FA), such as 18:5n-3, mainstream from the PKS pathway, might also be involved in toxic processes in Dinophytes [19,47,48]. It appears, then, that toxin production in harmful Dinophytes could be closely related to fatty acid synthesis.

Attempts to understand the synthesis pathway of microalgae FA mainly focused on identifying gene coding for the different enzymes (desaturase or elongase) involved in PUFA formation. For transcriptomic studies, they are challenging targets to enhance the production of 20:5n-3 and 22:6n-3 rich single-cell oil. This could help in counteracting the expected lack of these two healthy compounds in the future. Additionally, new isotopic techniques have been developed in recent years and present an important progress in understanding metabolic fluxes. The principle is to monitor the incorporation of an isotopic label, introduced as a substrate, into targeted organic macromolecules such as fatty acids [49,50,51,52] to quantify lipid production and identify synthesis pathways. This had already been applied in *E. coli* [53], yeast [52], and microalgae [33,54,55,56]. In compound-specific isotopic analysis (CSIA), these techniques couple gas chromatography with mass spectrometry (GC-c-IRMS) and allow the direct resolution of the isotopic composition of organic macromolecules, including fatty acids [57,58,59,60]. 

The present study aims to investigate the synthesis pathways of n-3 PUFAs by the toxic Dinophyte *Alexandrium minutum* using a stable isotope (^13^C) labelling experiment. This species is responsible for a harmful bloom (red tide) that can cause the severe mortality of marine organisms and has been extensively investigated. Three culture balloons followed; two replicates received ^13^CO_2_, while the last one was used as a control and was bubbled with unlabeled CO_2_ of petrochemical origin (depleted in ^13^C). The incorporation of the ^13^C was monitored in eleven FAs for 54 h at a high to moderate temporal resolution. The progressive incorporation of the ^13^C-labelled CO_2_ into FA (from precursors to n-3 PUFA) and quantification of main fatty acid production allowed us to constrain FAS, elongase/desaturase, as well as the PKS pathway and their involvement in PUFA production by *A. minutum*. In parallel with the ^13^C-monitoring, growth and other cellular parameters (morphology, viability, and lipid content) were monitored by flow cytometry analysis to assess the microalgae physiological status. 

## 2. Materials and Methods 

### 2.1. Algal Culture and Isotopic Labelling

This experiment was conducted following the experimental design described by Remize et al. [33]. Briefly, the toxic Dinophyte *A. minutum* (strain AM89BM, isolated from a bloom in the Bay of Morlay, France) were cultured in batch (total volume of 2 L) under continuous light (24 h light cycle, 100 µmoles photons m^−2^·s^−1^ at 20 °C). The culture was prepared with 1750 mL of sterile sea water, 250 mL of algal inoculum (pre-cultured for five days to reach the exponential growth phase), and 2 mL of nutrient medium (L1 medium) [61]. The culture was kept sterile during the entire experiment. Two replicate balloons received the ^13^C-label (named, later on, Alm1 for balloon 1 and Alm2 for balloon 2). A third balloon was used to check the absence of a negative impact of ^13^C-labelled CO_2_. This control balloon was prepared with petrochemical CO_2_ (non-^13^C labelled) but has not been analyzed further for fatty acid composition. 

The two replicates were subjected to a pre-culturing phase of four days before the start of the monitoring. The isotopic labelling was performed using pure ^13^C-CO_2_ gas (Sigma-Aldrich, Saint Louis, MO, USA, <3%atom 18O, 99.0%atom ^13^C). The CO_2_ introduction in the culture was started just before the first sampling time (t_0_) and was maintained for 54 h (t_54_). The growth was then controlled by ^13^C-CO_2_ addition using a pH-stat system, which supplied the culture when the pH was superior to 8.00 ± 0.05 (Figure 1). 

### 2.2. Sample Collection

Sampling was performed regularly for 54 h at the following time points: 0, 2.5, 5, 10, 20, 24, 30, 48, 54 h (nine samples per balloons). The experimental design was created to collect the culture medium without opening the balloon and to avoid bacterial contamination and the introduction of atmospheric CO_2_. At each sampling time, a total volume comprised of between 82 and 142 mL was collected for (i) the flow cytometry (FCM) analysis of cellular parameters, (ii) the isotopic analysis of particulate organic carbon (^13^C-POC) and dissolved inorganic carbon (^13^C-DIC) by EA-IRMS, (iii) fatty acid (FA) analysis in neutral lipids (NLs) and polar lipids (PL) by GC-FID, and (iv) the compound-specific isotope analysis (CSIA) of FA (^13^C-FA) by GC-c-IRMS, as described in the following paragraphs.

### 2.3. Flow Cytometry Analysis

Algae cellular variables were measured using an Easy-Cyte Plus 6HT flow cytometer (Guava Merck Millipore^®^, Darmstadt, Germany) equipped with a 488 nm blue laser; detectors of forward (FSC) and side (SSC) light scatters; and three fluorescence detectors: green (525/30 nm), yellow (583/26 nm), and red (680/30 nm). Cell variables—i.e., forward scatter (forward scatter, FSC), side scatter (side scatter, SSC), and red fluorescence (FL3, red emission filter long pass, 670 nm, a proxy of chlorophyll content)—were used to identify and select the *A. minutum* cell population. FSC and SSC give, respectively, information on the relative size and complexity of cells [62,63,64]. The flow cytometry measurements were performed on fresh (live) samples.

Two fluorescent probes were used to assess the viability and lipid content according to Lelong et al. (2011) and Seoane et al. [65]. SYTOX (Molecular Probes, Invitrogen, Eugene, OR, USA at a final concentration of 0.05 µM) was used to estimate the percentage of dead cells in the Dinophyte samples [66]. The BODIPY probe (BODIPY 505/515 FL; Molecular Probes, Invitrogen, Eugene, OR, USA, final concentration of 10 mM) was used as a proxy of the lipid reserves [65]. 

The concentration of bacteria was also monitored during the experiment according to Seoane et al. [65] using SYBR^®^Green staining (Molecular Probes, Invitrogen, Eugene, OR, USA, #S7563). The results are expressed as the concentration of bacteria per mL.

### 2.4. POC Concentration and Bulk Carbon Isotopic Composition

Samples (40–70 mL) for Particulate Organic Carbon (POC) concentration and stable isotopic composition were processed as described in Remize et al. [33]. The POC concentrations of all samples were measured using a CE Elantech NC2100 (ThermoScientific, Lakewood, NJ, USA) according to the protocol of the United States Environmental Protection Agency [67] with acetanilide (99.9% purity, C_8_H_19_NO CASRN 103-84-4) as a standard. The POC concentrations are given in mmol L^−1^. The bulk carbon isotopic composition (^13^C-POC) was analyzed by continuous flow on an Elemental Analyzer (EA, Flash 2000; Thermo Scientific, Bremen, Germany) coupled with a Delta V+ isotope ratio mass spectrometer (Thermo Scientific, Bremen, Germany). Calibration was performed with international standards and the in-house standards described in Table 1.

### 2.5. DIC Concentration and Bulk Carbon Isotopic Composition

Samples for the Dissolved Inorganic Carbon (DIC) concentration and stable isotopic composition were collected from the filtrate of POC samples and processed as described in Remize et al. [33] and following the protocol of Assayag et al. [68]. The DIC was measured using a gas bench coupled with a Delta Plus mass spectrometer from Thermo Scientific, Bremen, Germany (GB-IRMS). 

### 2.6. Isotopic Data Processing

We used the atomic abundance of ^13^C in percent (%atom of ^13^C) to express the results. Conversion between delta notation and %atom^13^C notation can be performed according to Larsson et al. [69]: (1)$${\%\mathrm{atom}}^{13}C=100\times \frac{\left(\frac{\delta {}^{13}C}{1000}+1\right)\times {\left(\frac{{}^{13}C}{{}^{12}C}\right)}_{VPDB}}{1+\left(\frac{\delta {}^{13}C}{1000}+1\right)\times {\left(\frac{{}^{13}C}{{}^{12}C}\right)}_{VPDB}},$$
where (^13^C/^12^C)_PDB_ = 0.0112372, the ratio of ^13^C to ^12^C in the international reference VPDB standard.

The atomic enrichment (AE) was then calculated from the atom %^13^C-POC and corrected with the values of the control balloon (POC_control_ = 1.08%) and atom %^13^C DIC corrected from the control balloon (DIC_control_ = 1.12%) according to the following equations: (2)$$AE_{POC}={\%atom}^{13}C-POC_{control},$$
(3)$$AE_{DIC}={\%atom}^{13}C-DIC_{control}.$$

### 2.7. Fatty Acid Analysis

#### 2.7.1. Lipid Extraction

After filtration on pre-combusted (450 °C, 6 h) 47 mm GF/F filters (porosity 0.7 µm) with 40–70 mL of culture and the addition of boiling water to stop the lipase activity, the lipids were extracted by diving the filter into a 6 mL solvent mixture (chloroform:methanol; 2:1 *v*/*v*). The lipid extracts were flushed with nitrogen and stored at −20 °C until analysis. 

#### 2.7.2. Separation of Neutral and Polar Lipids

The lipid extracts were separated into neutral and polar fractions following the method of Le Grand et al. [70] and as described previously in Remize et al. [33]. In brief, 1 mL of extract was evaporated to dryness with nitrogen, recovered with three washes of 0.5 mL of chloroform:methanol (98:2 *v*:*v*; final volume 1.5 mL), and spotted at the top of a silica gel column (40 mm × 4 mm, silica gel 60A 63–200 µm 70–230 mesh rehydrated with 6% H_2_O, Sigma-Aldrich, Darmstadt, Germany). The neutral lipid fraction (NL) was eluted using chloroform:methanol (98:2 *v*:*v*; 10 mL) and the polar lipid fraction (PL) with methanol (20 mL). Both were collected in glass vials containing an internal standard (C23:0 as free fatty acid, 2.3 µg). The lipid fractions were then stored at −20°C until further analysis. 

#### 2.7.3. Fatty Acid Analysis by Gas Chromatography (GC-FID and GC-c-IRMS)

Transesterification and fatty acids methyl esters (FAME) analysis with gas chromatography (GC-FID and GC-c-IRMS) were conducted according to the protocol described by Mathieu-Resuge et al. [71]. The analysis of the FAME was performed on a Varian CP8400 gas chromatograph (Agilent, Santa Clara, CA, USA) using simultaneously two parallel separations on polar (ZBWAX: 30 mm × 0.25 mm ID × 0.2 µm, Phenomenex, Torrance, CA, USA) and apolar columns (ZB5HT: 30 m × 0.25 mm ID × 0.2 µm, Phenomenex, Torrance, CA, USA). The FAME were identified by the comparison of their retention times with those of commercial and in-house standards mixtures, as shown for the two columns used in the Appendix A. The individual fatty acid concentrations obtained in µg·L^−1^ by GC-FID were also expressed in µmolC·L^−1^ (µg·L^−1^/molecular weight of individual fatty acid × carbon number of individual fatty acid) to ease the comparison with the POC concentrations expressed in µmolC L^−1^ as well. The total fatty acid concentrations in neutral lipids, polar lipids, and the sum of both fractions (the latter named hereafter TFA) were expressed in µmolC·L^−1^. 

The compound-specific isotope analyses (CSIA) of FAME were performed on a Thermo Fisher Scientific GC ISOLINK TRACE ULTRA (Bremen, Germany) using the same apolar column as for the FAME analysis. Only the 11 fatty acids with the highest concentrations (> 1 µmolC·L ^−1^) as measured by the GC-FID analyses were considered for CSIA, as they presented a signal amplitude on the GC-c-IRMS superior to the threshold value of 800 mV defined by Mathieu-Resuge et al. [71]. The minor FA presenting a too-low signal amplitude (<800 mV) on the GC-c-IRMS did not allow precise isotope ratio analysis. 18:1n-9 and 18:3n-3 were co-eluted for GC-c-IRMS on the apolar column so they were considered together. Examples of GC-c-IRMS chromatograms for C_18_ PUFAs 18:4n-3 and 18:5n-3 as well as the confirmation of their identification by GC-MS are available in the Appendix A.

The FA atomic enrichment (AE_FA_) was estimated with the same method as for AE_POC_ and AE_DIC_, with the %atom^13^C for each FA given by GC-c-IRMS analysis (AE_FAcontrol_ = AE_FAnat_ = 1.08%). 

#### 2.7.4. Compound Specific Isotope Data Processing 

To evidence the FA conversion of fatty acid A into fatty acid B in *A. minutum*, we calculated the AE_FA_ ratio (R) of product B over hypothesized precursor A. If the calculated ratio was close to one with a confidence interval calculated at α = 0.1 (defined arbitrarily), the fatty acids A and B were supposed at the equilibrium in terms of the label incorporated and can be assumed to be synthesized simultaneously or very closely. If the ratio was below one, the transformation of A into B was possible but slower. On the contrary, if the ratio was above one, A could not be considered as a major precursor of B. R was defined as follows.
(4)$$R=\frac{AE_{FA\left(B\right)}}{AE_{FA\left(A\right)}},$$
where A the fatty acid is supposed to be a precursor of fatty acid B and AE_FA(A)_ and AE_FA(B)_ are their respective atomic enrichments at each sampling time.

### 2.8. Statistical Analysis

To assess the potential effect of the time and difference between balloons during algae development and ^13^CO_2_ incorporation, a PERMANOVA analysis was used on the FA percentage separately in NL and PL. The Spearman test was conducted on the fatty acid abundance in both PL and NL to explore the correlations between individual fatty acids.

## 3. Results

### 3.1. Algal Physiology during Growth

*A. minutum* showed a large lag phase before increasing at t_30_ (Figure 2a) with oscillating dynamics: an increase between t_2.5_ and t_20_ and a decrease in algal abundance between t_20_ and t_30_ (mean abundance decreased from 1.2 × 10^4^ cells·mL^−1^ to 7.4 × 10^3^ cells·mL^−1^). Between t_30_ and t_54_, the algal abundance was multiplied by a factor of five (7.4 × 10^3^ to 36.4 × 10^3^ cells·mL^−1^). The bacterial abundance presented a general increase from t_0_ to t_54_ (Figure 2b). The decrease in algal abundance between t_20_ and t_30_ was not observed with bacterial abundance.

The cell size and complexity, respectively, estimated by FSC and SSC remained relatively stable for 48 h (FSC = 160 ± 37 a.u, SSC = 671 ± 8 a.u). The FSC increased slightly after t_48_ (181 ± 5 a.u), while the SSC decreased (602 ± 6 a.u). The neutral lipid content (BODIPY, green fluorescence) increased until t_10_ (107 ± 18 to 158 ± 25 u.a), decreased until t_48_ (158 ± 25 to 58 ± 9 a.u.), and increased again thereafter (58 ± 9 to 81 ± 3 a.u). The chlorophyll content (FL3, red fluorescence) increased between t_0_ and t_24_ (547 ± 5 to 606 ± 12 a.u) and decreased after t_24_ (606 ± 12 to 522 ± 2 a.u). The viability, as monitored by SYTOX green fluorescence, stayed stable during the whole experiment (Table 2).

### 3.2. POC and TFA Concentrations

The Particulate Organic Carbon (POC) concentration followed an increasing trend from t_0_ to t_54_ for *A. minutum* (R^2^ = 0.94, *p*-value < 0.05). The concentrations were similar between the two balloons and augmented from 1.1 ± 0.2 to 2.9 ± 0.1 mmolC L^−1^ (Figure 3a). The Total Fatty Acids (TFA) concentration remained relatively stable between t_0_ and t_30_ (270 ± 5 µmolC L^−1^) and decreased after t_48_ and t_54_ (180 ± 3 µmolC L^−1^) (Figure 3b). 

### 3.3. Bulk POC and DIC and Their ^13^C-Labelling

The Dissolved Inorganic Carbon (DIC) and POC atomic enrichments with time are shown in Figure 4. The DIC atomic enrichment increased quickly between t_0_ and t_10_ (from 0 to 50%), remained stable between t_10_ and t_30_ (around 50%), and increased to around 60% at t_48_ and t_54_. The two balloons had a similar DIC enrichment. On the contrary, the POC atomic enrichment was continuous but slower from t_0_ to t_54_. The Alm1 had a slightly higher POC enrichment after t_30_. The final POC enrichments for Alm1 and Alm2 were 58% and 45%, respectively.

### 3.4. Bulk POC and DIC and Their ^13^C-Labelling

Overall, there was more PL (61 ± 11%) than NL (39 ± 11%) (Figure 5b), but their respective proportions varied greatly during the culture; the percentage of NL over TL decreased from 56% to 25% between t_0_ and t_30_ and then increased back to about 60% at t_54_ (Figure 5c). 

For *A. minutum*, 36 FAs were identified but 24 were present in a trace amount (<1% of TFA). The 12 main FAs were the ones presented in Figure 5a. Appendix A gives the concentration in µg L^−1^ and µmolC L^−1^ for all the fatty acids identified in this study. The PERMANOVA conducted on PL and NL FA showed a significant difference between the sampling times for the fatty acid percentages (*p*-value < 0.05), but no significant difference was observed between balloons. The replicates could then be considered together. The SFAs and PUFAs were homogenously represented in the NL (40% each, 20% for MUFA), and while in the PL PUFA was the main category of FA (63% against 32% for SFA and only 5% for MUFA). 16:0, 18:1n-9, 14:0, 18:2n-6, and 18:3n-3 were the main FAs of the NL fraction (respectively, 27 ± 2%, 19 ± 5%, 10 ± 0.004%, 10 ± 1%, and 9 ± 1%). For the PL fraction, 16:0 (23 ± 8%) and 22:6n-3 (20 ± 3%) were the most abundant FA, followed by 18:4n-3 (15 ± 4%) (Figure 5a). At t_48_, in the NL fraction the PUFA and SFA proportions were augmented (39% to 43% for both), while in PL the PUFA decreased (68% between t_0_ and t_30_ and 50% after t_30_) (data not shown). The MUFA decreased in NL (23% between t_0_ and t_30_ and 11% after t_30_) and remained relatively stable in PL. The n-3 PUFAs were always more important than the n-6 PUFAs in both NLs (28% vs. 11%, respectively) and in PL (51% vs. 11%). The bacterial fatty acids (including branched, C_15_, C_17_ and C_21_ FA) of the *A. minutum*-associated bacteria communities (non-axenic culture) remained below 0.8% in the NL and PL fractions and were not further followed. 

### 3.5. Relationship between FA

Following the Spearman correlation test, significant (*p*-value < 0.01) positive correlation factors (ρ) were found between fatty acids for 18:3n-3 and 18:4n-3 in both PL and NL (ρ = 0.67 and 0.74, respectively), 18:5n-3 and 18:4n-3 in PL and NL (ρ = 0.50 and 0.48, respectively), and 20:5n-3 and 22:6n-3 in PL and NL (ρ = 0.68 and 0.92, respectively). 22:5n-3 was correlated with 20:5n-3 (ρ = 0.54) and 22:6n-3 (ρ = 0.52) only in the NL fraction and with 18:2n-6 to 18:3n-3 also only in the NL s(ρ = 0.91).

### 3.6. Fatty Acids ^13^C-Enrichment 

The atomic enrichments (AE) with time of the 11 FAs analyzed by GC-c-IRMS are presented in Figure 6. All the studied FAs appeared to have a similar continuous enrichment to that of POC. The AE of 18:1n-9+18:3n-3 and 18:2n-6 in the NL fraction was saturated at t_54_ with an enrichment level superior to AE_DIC_ (56%). All the FAs presented an increasing ^13^C level from t_0_ to t_54_. The enrichment dynamics were slightly different between t_0_ to t_30_ (phase I) and t_48_ and t_54_ (phase II). 

During phase I, 22:6n-3 was always the most enriched fatty acid in both the NL and PL fractions (35 ± 0% and 32 ± 0% at t_30_, respectively). It was closely followed by 22:5n-3 in both fractions (31 ± 1% and 25 ± 2% at t_30_, respectively). The order of enrichment, from the most to the least enriched FA, during this first phase for NL was the following: 22:6n-3, 22:5n-3, 18:2n-6, 16:0, 18:1n-9+18:3n-3, 20:5n-3, 18:4n-3, 18:5n-3, 22:5n-6, and 18:0. In the PL fraction, the order was similar, with only 22:5n-6 being more enriched than 18:4n-3, 18:5n-3, and 18:0. 

After t_30_ (Phase II), 18:1n-9+18:3n-3 and 18:2n-6 in the NL fraction became the most enriched fatty acids (both were saturated in ^13^C starting t_48_ AE_FA_ > AE_DIC_), followed by 22:6n-3 (56 ± 1% at t_54_). The enrichment succession of other fatty acids in the NL after t_30_ was the following: 16:0, 22:5n-3, 18:4n-3, 20:5n-3, 18:5n-3, 22:5n-6, and 18:0. For PL after t_30_ and until the end of the monitoring, 22:6n-3, 18:1n-9+18:3n-3, 18:2n-6, and 22:5n-6 were the most enriched FAs (above 50% for 22:6n-3 and 18:1n-9+18:3n-3, and above 45% for 18:2n-6 and 22:5n-6 at t_54_). During this second period, 22:5n-6 and 18:2n-6 presented similar enrichment levels (Figure 6). The other FAs were decreasingly enriched as follows: 16:0, 22:5n-3, 20:5n-3, 18:5n-3, 18:4n-3, and 18:0. 

Without considering the phases, the enrichment in NLs was always higher than the enrichment in PL for 16:0, 18:1n-9+18:3n-3, 18:2n-6, 18:4n-3, 18:5n-3, 20:5n-3, and 22:5n-3. For 22:5n-6 and 18:0, the enrichment was higher in the PL fraction during phase II (at t_54_, 46 ± 0% in PL vs. 29 ± 0% in NL for 22:5n-6; 21 ± 1% in PL vs. 9 ± 0% in NL for 18:0). 22:6n-3 was consistently presenting similar levels of enrichment in both the NL and PL fractions (Figure 6).

Table 3 explores the conversion between FA (hypothesized product versus precursor) according to their respective AE. All the studied ratios were superior to one exception made for 18:5n-3/18:4n-3 in NL. 

## 4. Discussion

This experiment investigated the ^13^CO_2_ incorporation into fatty acids to understand how n-3 long-chain polyunsaturated fatty acids were synthesized by the Dinophyte *Alexandrium minutum*. Both DIC and POC were enriched rapidly after only five hours of incubation, and this was before the algae reached its exponential growth phase at t_30_. The algae physiology (cell size, cell complexity, and viability) was not negatively impacted by the ^13^CO_2_ bubbling and remained stable during the experiment. However, the neutral lipid and chlorophyll contents appeared to undergo some modification during the 54 h of monitoring and will be discussed in the next sections.

The lipid profile of *A. minutum* was close to that of the other Dinophytes; it was rich in 18:4n-3, 18:5n-3, 20:5n-3, and 22:6n-3 [72,73,74]. In our experiment, the n-3 PUFAs were the most abundant. 

Among the n-3 PUFAs, 22:6n-3 was one of the most produced and the most enriched during the entire monitoring. It was more enriched than other FAs, such as 18:4n-3, 18:5n-3, 20:5n-3, or 22:5n-3, which were so far considered as its potential intermediates in the n-3 “conventional” synthesis pathway [19]. Moreover, the synthesis of 22:6n-3 from 22:5n-3 via a Δ4 desaturase of the n-3 pathway or from 22:5n-6 via a Δ19 desaturase of the ω3-desaturase pathway was unlikely, as the atomic enrichment ratios of 22:6n-3/22:5n-3 and 22:6n-3/22:5n-3 were both above one. Therefore, we assumed that none of these two routes could be the main source of 22:6n-3 in *A. minutum*. 

The O_2_-independent PKS pathway was identified in different microalgae taxa [18,30]. It relies on the same four basic reactions as the FAS pathway (condensation, reduction, dehydration, reduction) and is responsible for long-chain polyunsaturated fatty acids synthesis, such as 20:5n-3 and 22:6n-3 [19,75]. Contrarily to the FAS pathway, the PKS pathway is less energy-consuming because it requires fewer reduction and dehydration steps [18]. Identifying the FA precursors of this pathway is difficult, as these intermediates might not be released from the enzyme protein into the fatty acids pool and would therefore be present in very small concentrations [19]. Several PKS genes have been identified in microalgae, including Dinophytes, in recent years [25,26,30]. Regarding the rapidity of 22:6n-3 synthesis and the absence of more enriched precursors during the monitoring, we suggested that 22:6n-3 was produced directly from the PKS pathway in *A. minutum* (Figure 7). Moreover, because the enrichment was similar in both the NL and PL fractions, we proposed that the PKS pathway in this Dinophyte might exist to produce 22:6n-3 in both membrane lipids (PL) and reserve lipids (NL). 

During our monitoring, *A. minutum* exponential growth started only after 30 h and resulted in an eight-fold increase in cell abundance. Before this growing phase, despite presenting a stable viability the algae underwent modifications of their cellular content with a decrease in the chlorophyll content (FL3) and neutral lipid content (BODIPY). The diminution of the chlorophyll content might be an indicator of reduced photosynthesis following a stress [76,77], while the diminution of neutral lipid content would reflect a remobilization of the stored lipids. This could be associated with encystment [42]. Indeed, Dinophyte species are able to form temporary cysts in response to unfavorable environments. Temporary cysts enable functioning over shorter time scales to answer the perturbation of environmental conditions [42]. The formation of temporary cysts can be associated with the repression of photosynthesis activity (reduction in chlorophyll concentration) [42]. During temporary encystment, the cysts rely on their reserves, which can also explain why we observed a decrease in the NL concentration before t_30_ and a slight decrease in the TFA content after t_30_ in *A. minutum*. 

In parallel with the loss of neutral lipids and decrease in TFA concentrations, most of the FAs were nevertheless enriched in priority in the NL fraction, especially 22:6n-3 and other n-3 PUFAs (18:4n-3, 18:5n-3, 20:5n-3, 22:5n-3). However, the succession of FA atomic enrichment into fatty acids and the atomic enrichment ratios of these n-3 PUFAs did not support the existence of an active n-3 pathway to produce C_20_ and C_22_ PUFAs from their potential C_18_ PUFAs precursors. Indeed, 18:4n-3 and 18:5n-3 were enriched after C_20_ and C_22_ PUFAs. Furthermore, the 22:5n-3/20:5n-3 and 22:6n-3/22:5n-3 ratios were >1, implying that 20:5n-3 and 22:5n-3 cannot be the respective precursors of 22:5n-3 and 22:6n-3. Thus, it seemed improbable that these C_20_ and C_22_ PUFAs were synthesized by the known “conventional” n-3 pathway. Two explanations can be proposed. First, as for 22:6n-3, 20:5n-3 might have been produced from a different and independent PKS pathway (Figure 7). This implies that PKS enzymes do not release their intermediates, but only the final PUFA end product, 20:5n-3 or 22:6n-3. The absence of 16:4n-3, a mainstream PKS pathway intermediate [19], had already been reported in Dinophytes by Leblond and Lasiter [78]. Furthermore, Jang et al. [37] have demonstrated that PKS genes were upregulated during the formation of resting cysts in Dinophytes. This could also be the case in temporary cysts, and this then would support the use of a PKS pathway to produce LC-PUFAs without O_2_ during encystment. Based on the enrichment dynamics, 22:6n-3 synthesis by PKS would be faster than the hypothesized parallel PKS pathway for 20:5n-3. The existence of such a faster route to produce 22:6n-3 in Dinophytes might also explain why, in some species such as *Crypthecodinium cohnii*, putative n-3 PUFAs intermediates were only identified in trace amounts [79]. *C. cohnii*, an exclusive heterotroph Dinophyte, was previously speculated to produce 22:6n-3 primarily by the PKS pathway [80]. We can then assume that the 22:6n-3 PKS pathway could be active in both heterotroph and mixotroph Dinophytes. However, it might not be the case for the 20:5n-3 PKS routes, as *C. cohnii* did not produce it in significant quantities [79].

The degradation of PUFA by β-oxidation coupled with fatty acid remodeling by lipase could also explain the decreasing order of n-3 PUFA enrichment, from a longer to a shorter chain length, as observed in *A. minutum*. As suspected earlier, it appeared that quiescent cells repressed their photosynthetic activities during temporary encystment [42]. However, energy metabolism, glycolysis, and even β-oxidation remain active [44]. β-oxidation is the principal pathway implicated in FA degradation and occurs in the peroxisome of plants and microalgae [81]. It relies on four enzymatic reactions (oxidation, hydration, dehydrogenation, and thiolytic cleavage) to form the end product acetyl-CoA [81]. The LC PUFA 22:6n-3 is the most abundant fatty acid found in the triacylglycerol (TAG) of Dinophytes [82] and is found in large proportion in phospholipids [74]. Enzymes responsible for the cleavage of reserve lipids (NL) or membrane lipids (PL) could release 22:6n-3 from the glycerol backbone and then join the free fatty acids (FFA) pool to be degraded by β-oxidation in the peroxisomes. PUFAs such as 20:5n-3 and 18:5n-3 are the intermediates of 22:6n-3 β-oxidation. They can be further degraded into acetyl-CoA, or serve as an acyl-donor for the de novo synthesis of TAG. This hypothesis would explain why all the n-3 PUFAs were primarily enriched in the NL fraction. The remodeling of membrane lipids during encystment had already been observed by Lichtlé and Dubacq [83]. Moreover, Xu et al. [84] reported that the 20:5n-3 issued from 22:6n-3 β-oxidation was recycled into TAG in yeast after the expression of type-2 diacylglycerol acyltransferase (DGAT2) from *Thalassiosira pseudonana*. Their study provided evidence that DGAT2 could be involved in both the de novo synthesis of TAG as well as the TAG remodeling with β-oxidation intermediates. Dinophytes were reported to have type 2 DGAT [85]. However, the enzymes (lipases) involved in the cleavage of TAG and membrane lipids as well as β-oxidation processes remain to be fully investigated in Dinophytes.

According to the atomic enrichment, 18:4n-3 was synthesized after all the long-chain n-3 PUFAs. It was likely synthesized from 18:3n-3 by the conventional n-3 pathway for both the PL and NL, fractions even if we were not able to prove it with ratio calculation as 18:3n-3 and 18:1n-9 coeluted. Δ6 desaturase involved in the desaturation of 18:3n-3 into 18:4n-3 has been identified in the Dinophyte *Lingulodinium polyedrum*, allowing the production of 18:4n-3-rich galactolipids [86]. The ratio 18:5n-3/18:4n-3 was below one in NL (<0.55), indicating a feasible transformation of 18:4n-3 into 18:5n-3, although this ratio was above one in PL (>1.67). The 18:5n-3 could be formed from 18:4n-3 in NL by the action of Δ3 desaturase, as suggested by Joseph [87] to explain the presence of this unusual FA in Dinophytes. The 18:5n-3 could also be synthesized by the PKS pathway in the PL fraction, since it had been previously identified as a mainstream PKS fatty acid [19] (Figure 8). C_18_ PUFAs, especially 18:4n-3 and 18:5n-3, seemed to be final products in *A. minutum* and not be further desaturated, as they presented some of the lowest enrichments in our experiment. 18:4n-3 and 18:5n-3 have been reported to be the dominant fatty acids of Dinophyte glycolipids, especially MGDG and DGDG [78,88]. In *A. minutum*, 18:4n-3 and 18:5n-3 might have a similar location. 18:5n-3 and 18:4n-3 have been also described to be present in free fatty acids, revealing a higher toxicity compared to that of other PUFAs in Dinophytes [47,48]. They could then be involved in *A. minutum* toxicity.

18:2n-6 was always more enriched than the other n-6 PUFA, 22:5n-6, in NL. Their enrichments were even similar in the PL. 18:2n-6 and 22:5n-6 are both linked in the classical n-6 pathway [19]. Based on the atomic enrichments, 18:2n-6 could be initially produced in the NL fraction and then transported to the membranes (PL). Řezanka et al. [82] showed than 18:2n-6 and 16:0 were mainly found in phospholipids (PCs) in Dinophytes. Even though the enrichment of the known precursors of 22:5n-6 cannot be quantified due to their low concentrations, 18:2n-6 could be introduced into PC to be furtherly desaturated into 22:5n-6. Finally, 22:5n-6 could be integrated into TAG. Whether PC is involved in acyl-editing mechanisms during TAG synthesis remains to be described in a Dinophyte. Another hypothesis regarding 22:5n-6 synthesis could be the existence of a n-6 PKS pathway.

After t_30_, the algae entered into a vegetative phase and grew exponentially. It was associated with an increase in the SFA and PUFA proportion in the NL and a decrease in the PUFA proportion in the PL. TAGs are useful to exit quiescence and facilitate the initiation of the vegetative phase in Dinophytes [89]. 18:1n-9+18:3n-3 and 18:2n-6 became more enriched than 22:6n-3 in the NL after t_30_. The 18:1n-9 produced by the FAS pathway can be used in the de novo TAG synthesis pathway, called the Kennedy pathway, and be acylated in the positions *sn*-1 and *sn*-2 of DAG [90,91]. We assumed that de novo TAG synthesis has probably been reinitiated in *A. minutum*, and that it was directly driven to reserve lipid synthesis to compensate for those lost during excystment (increase in NL proportion and concentration after t_30_). 

## 5. Conclusions

The fatty acid synthesis in the Dinophyte *Alexandrium minutum* went through different routes. The PKS pathway appeared to be a particularly fast synthetic process, responsible for the high enrichment and production of the polyunsaturated fatty acid 22:6n-3. 22:6n-3 seemed to have a central role in maintaining a good physiological state, including during encystment. In our study, we assumed that the lag phase observed reflected the possibility of temporary encystment in *A. minutum*. It was characterized by a significant decrease in the proportion of neutral lipids, corresponding to the consumption of the reserve lipids. The 22:6n-3 might be degraded during this process, and its degradation products might be involved in the re-synthesis of triacylglycerol during algae excystment. The enrichment dynamics of C18 PUFAs revealed that they are unlikely to be involved in the further desaturation and elongation steps of n-3 C_20_-C_22_ PUFAs. They appeared to be the final products of the classical n-3 pathway. The 18:5n-3 atomic enrichment makes possible its origin from the desaturation of 18:4n-3, the degradation of longer fatty acids such as 20:5n-3, or the PKS pathway. These C_18_ PUFAs have been assumed to play some role in *A. minutum*, such as in its toxicity. Further studies are needed to better constrain the PUFA synthesis pathway in *A. minutum*, and especially to further demonstrate the involvement of the PKS pathway following 22:6n-3 and 20:5n-3 synthesis.

## Figures and Tables

**Figure 1 biomolecules-10-01428-f001:**
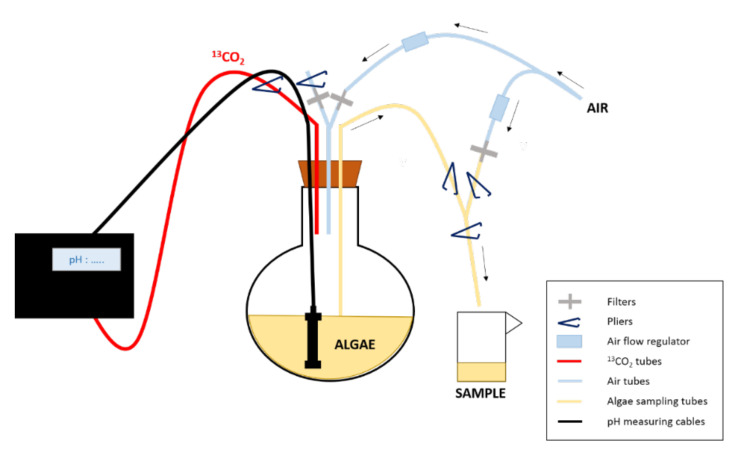
Experimental design of the enrichment experiment. The ^13^CO_2_ is supplied to the culture depending on its pH. To sample the algae, pliers are used to close/open the tubes/ways needed to first put the balloon under pressure and then allow sampling and finally rinse the tubes after sampling.

**Figure 2 biomolecules-10-01428-f002:**
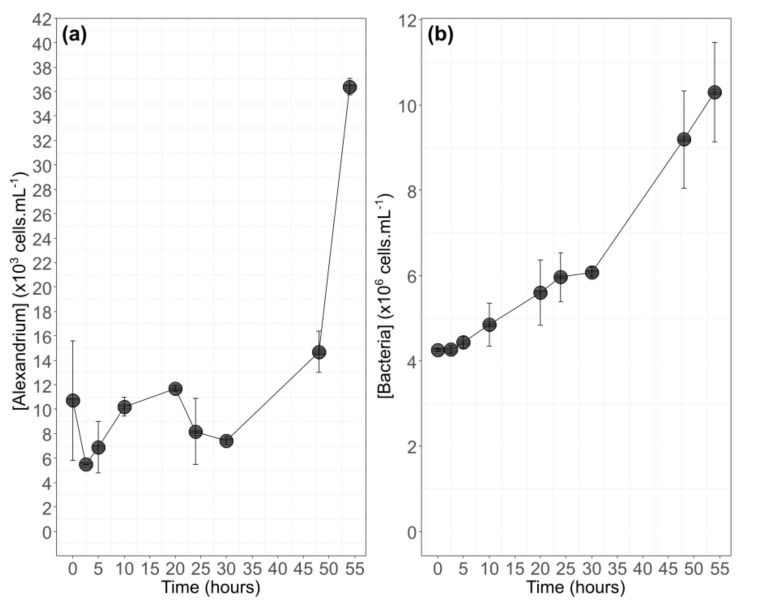
Mean cell concentration (mean ± SD) of the 2 enriched balloons of *A. minutum* (**a**) and the corresponding bacteria concentration during the 54 h of the experiment (**b**).

**Figure 3 biomolecules-10-01428-f003:**
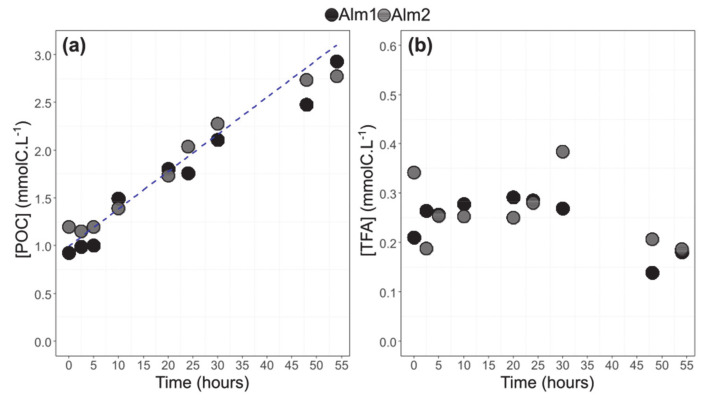
Particulate organic carbon concentration (**a**) and Total Fatty Acid (TFA) concentration (**b**). The blue line corresponds to a significant increasing trend between t_0_ and t_54_ (Alm1, Alm2, *A. minutum* cultures in balloons 1 and 2).

**Figure 4 biomolecules-10-01428-f004:**
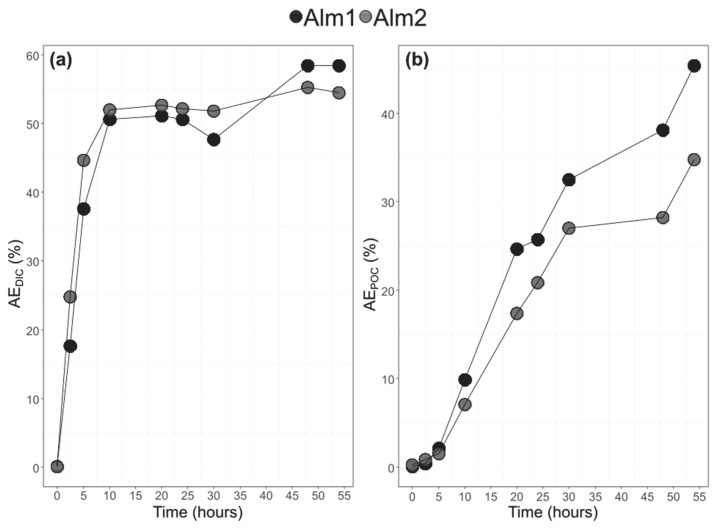
Atomic enrichment of the dissolved inorganic carbon (DIC) (**a**). Atomic enrichment of particulate organic carbon (POC) (**b**) (Alm1, Alm2, *A. minutum* cultures in balloons 1 and 2).

**Figure 5 biomolecules-10-01428-f005:**
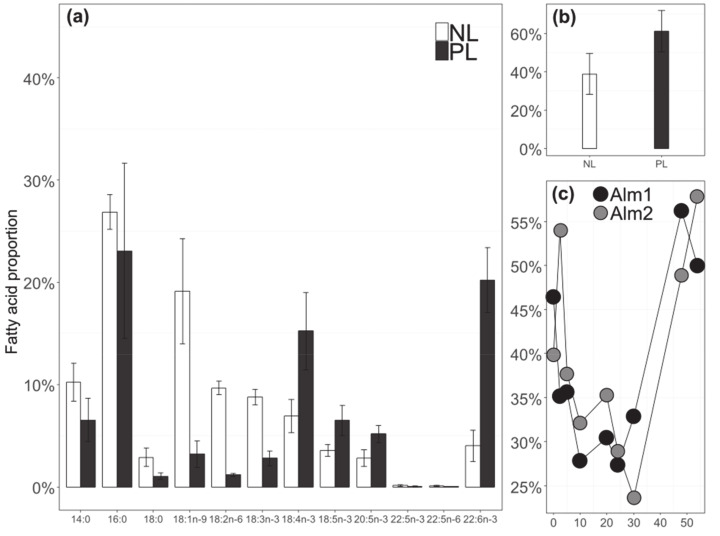
Proportions of the 12 main fatty acids of *A. minutum* (mean ± SD of the two enriched balloons over the studied time period between t_0_ and t_54_) (**a**) Mean proportion of neutral lipids (NL) and polar lipids (PL) during the experiment. (**b**) Variation in NL proportion with time for the two enriched balloons (**c**).

**Figure 6 biomolecules-10-01428-f006:**
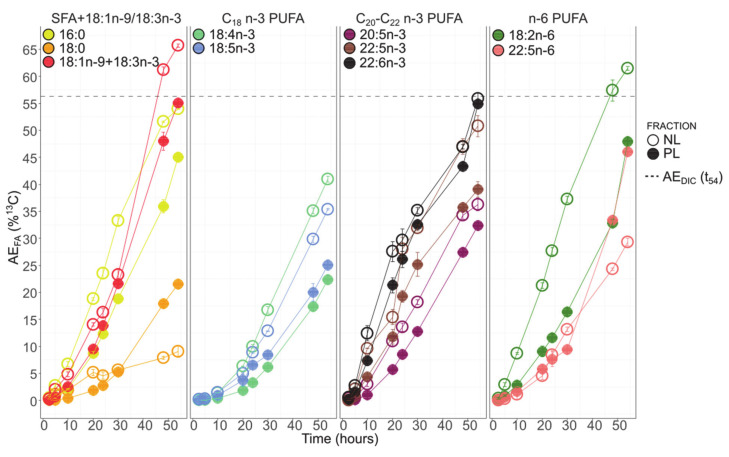
Atomic enrichment for the 11 main fatty acids (mean ± SD for the two enriched balloons). 14:0 was not considered as it was not involved in the studied fatty acids synthesis pathways to produce n-3 PUFAs. t_0_ is not shown here because *A. minutum* was not enriched at that time and t_2.5_ was kept as the reference. The dashed line represents the DIC atomic enrichment (AE_DIC_) at t_54_.

**Figure 7 biomolecules-10-01428-f007:**
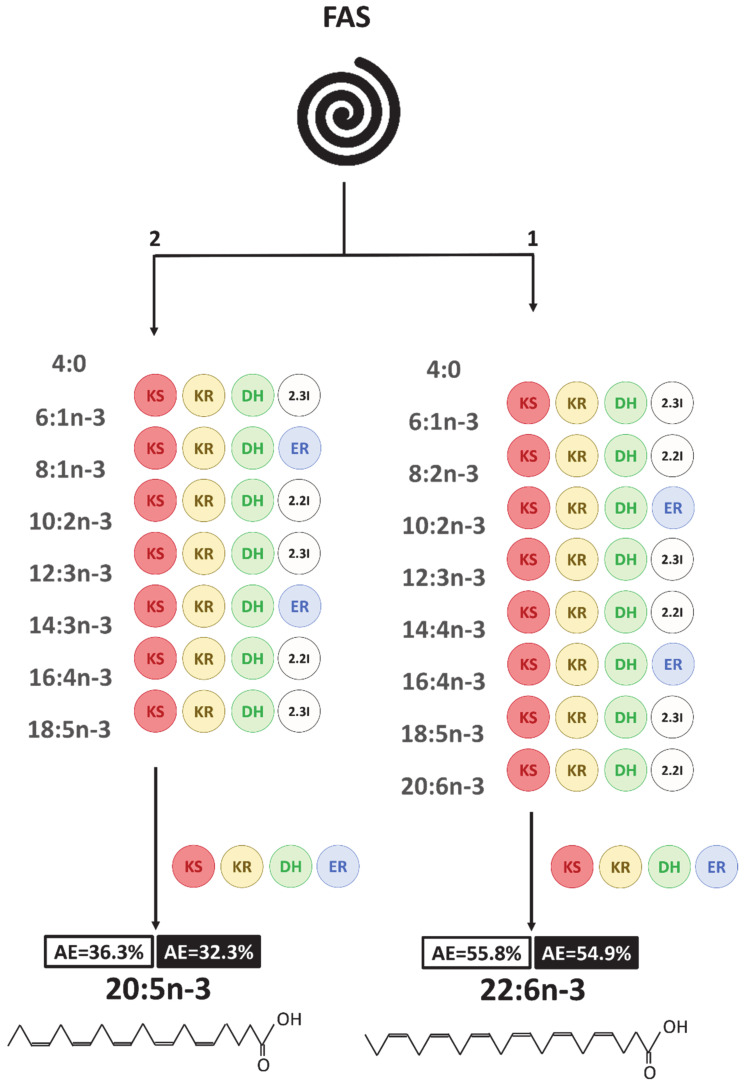
Hypothesized polyketide synthase (PKS) pathways in *A. minutum* to synthesize 20:5n-3 and 22:6n-3. Numbers in boxes correspond to the final AE value (at t_54_), white corresponds to neutral lipids and black to polar lipids. The PKS intermediates written in grey are not detected during the study, as the enzyme proteins might not release them until the final product is synthesized [19]. The circles represent the enzymes involved in these pathways: KS: 3-ketoacyl synthase; KR: 3-ketoacyl-ACP-reductase; DH: dehydrase; 2.2I/2.3I: 2-trans, 2-cis, or 2-trans, 3-cis isomerase; ER: enoyl-reductase. The numbers at the top of each pathways represent the suspected order of 20:5n-3 and 22:6n-3 synthesis according to the ^13^C-enrichment dynamics observed in this study.

**Figure 8 biomolecules-10-01428-f008:**
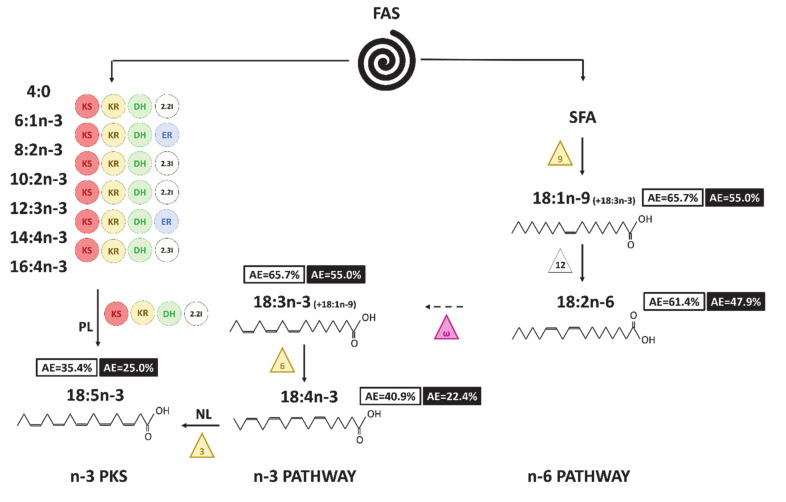
Hypothesized pathways to produce C_18_ FA in *A. minutum*. Numbers in boxes correspond to the final AE value (white for neutral lipids and black for polar lipids). The triangles symbolize the desaturases (front-end in yellow and methyl-end in purple), the circles the enzymes involved in the PKS pathway (KS: 3-ketoacyl synthase; KR: 3-ketoacyl-ACP-reductase; DH: dehydrase; 2.2I: 2-trans, 2-cis isomerase; 2.3I: 2-trans, 3-cis isomerase; ER: enoyl reductase). The ways with dashed arrows appear to be unlikely or cannot be proven with the enrichment dynamics. NL and PL written on the routes allowing the synthesis of 18:5n-3 indicate the fraction considered for each pathway.

**Table 1 biomolecules-10-01428-t001:** List of international and in-house standards used for the EA-IRMS and GB-IRMS analyses.

Description	Nature	Analysis	δ^13^C (‰)	SD
IAEA-CH_6_	Sucrose (C_12_H_22_O_11_)	^13^C-POC	−10.45	0.03
IAEA-600	Caffeine (C_8_H_10_N_4_O_2_)	^13^C-POC	−27.77	0.04
Acetanilide	Acetanilide (C_8_H_9_NO)	^13^C-POC	+29.53	0.01
CA21 (in-house std)	Calcium carbonate (CaCO_3_)	^13^C-DIC	+1476	
Na_2_CO_3_ (in-house std)	Sodium carbonate	^13^C-DIC	−6.8805	
NaHCO_3_ (in-house std)	Sodium bicarbonate	^13^C-DIC	−5.9325	

**Table 2 biomolecules-10-01428-t002:** Cellular parameters of *A. minutum*: morphology Side Scatter (SSC) and Forward Scatter (FSC)), viability (FL1-SYTOX), chlorophyll content (FL3), and neutral lipid content (FL1-BODIPY) using flow cytometry analysis (mean ± SD). Values for SYTOX are in % of total cells, and the values for BODIPY/FL3/SSC/FSC are in arbitrary units (a.u).

(h)	FL1-SYTOX ALIVE			FL1-BODIPY			FL3			SSC			FSC		
0	100	±	0	107	±	18	547	±	5	627	±	13	164	±	5
2.5	100	±	0	115	±	11	549	±	17	639	±	8	161	±	5
5	99	±	1	118	±	3	574	±	25	639	±	7	162	±	0
10	100	±	0	158	±	25	599	±	9	676	±	0	164	±	1
20	100	±	0	130	±	22	601	±	15	653	±	15	141	±	11
24	100	±	0	90	±	7	606	±	12	730	±	22	159	±	7
30	100	±	1	96	±	10	534	±	3	707	±	26	167	±	5
48	100	±	0	58	±	9	554	±	14	697	±	15	162	±	1
54	100	±	0	81	±	3	522	±	2	602	±	6	181	±	5

**Table 3 biomolecules-10-01428-t003:** Study of the FA synthesis pathways with the atomic enrichment (AE) ratios. Fatty acid A was assumed to be a major precursor of fatty acid B when the ratio B/A was equal or below the threshold value of one.

Fatty acid B/Fatty acid A	Neutral Lipids				Polar Lipids			
	Alm1		Alm2		Alm1		Alm2	
	Mean	SD	Mean	SD	Mean	SD	Mean	SD
18:5n-3/18:4n-3	0.56	0.16	0.53	0.15	1.67	0.64	1.70	0.55
20:5n-3/18:5n-3	1.28	0.38	1.48	0.61	2.04	0.56	2.20	0.59
22:5n-3/20:5n-3	1.94	0.85	1.81	0.62	2.24	1.32	2.15	1.00
22:6n-3/20:5n-3	2.32	1.04	2.25	0.99	3.33	2.15	3.32	2.21
22:6n-3/22:5n-3	1.21	0.30	1.23	0.29	1.45	0.23	1.47	0.29
22:6n-3/22:5n-6 *	3.22	1.87	3.89	1.61	2.74	1.27	3.15	1.51

Mean ratio values are considered here over the whole enrichment period between t_10_ and t_54_ for all the FAs, except (*) for 22:6n-3/22:5n-6 NL, which was averaged between t_20_ and t_54_.

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
