# Peer review of "Identification of Polyunsaturated Fatty Acids Synthesis Pathways in the Toxic Dinophyte Alexandrium minutum Using 13C-Labelling"

_biomolecules, 2020, doi:10.3390/biom10101428_

Round 1

Reviewer 1 Report

          My opinion about this MS is positive. Authors applied relatively new method for FA biosynthesis based on use stable 13C precursors, in this case 13CO2, and analysis of the processes by GC-MS.  Paper is continuation of study FA biochemistry in important marine phytoplankton. Very interesting is study progressive accumulation of label in FA during exponential growth phase. Experimental methods, results discussion are good. Based on 13C atomic enrichment (AE) and ratio AE supposed precursor/product were suggested biosynthesis ways for most essential for Dinophytes eicosapentaenoic (EPA), docosahexaenoic (DHA) and octadecapentaenoic (OPA) acids.  For EPA and DHA suggested two independent polyketide synthase (PKS) pathway. For production of acid 18:5(n-3) authors proposed “conventional” way with the use unusual D3 desaturase.

          Most of the results are confirm data on FA biosynthesis in microalgae, which also was not always direct.  At the same time this paper collected full information on lipid biochemistry in Dinophytes

  1. Why only 2 experimental sets were done?
  2. Why was used continuous light incubation?
  3. Some intermediates of FA biosynthesis may presents only as trace components, but they also can participate in metabolism.
  4. 6 not so convenient for accepting. Probably better put FA names into proper place in figure.
  5. I suggest add to results figure with mass spectra labeled fatty acids.
  6. Please add to Table 3 explanation of (mean ± SD) in this case.

Reviewer 2 Report

The manuscript is interesting but the hypothesis presented should be better discussed. There are differences of fatty acid concentrations between Neutral and Polar lipids and differences among the sampling times. The C18:4-n3 is high as C22:6n-3 in the polar lipids (Table Supplementary material).

The authors showed results of A.minutum concentration (Fig 2) and corresponding bacteria concentration without explanation. The main fatty acids from A.minutum were described (lines 288-298) but the bacterial branched fatty acids were reported. The authors should explain if bacteria strains were present in the balloons.

The analysis of fatty acids was poorly described. The standards used should be informed as well as for C18:5n-3, because Supelco 37 and  PUFA standards (Supplementary material) did not  contain C18:5. The description of the figure 5 is unclear for the fatty acid concentration (%) and no sampling time was supplied. The concentration of fatty acids (Table Supplementary material) is different.

The authors should explain how the identification of C18:5n-3 was performed. This is an uncommon fatty acid and the results should be clearly presented. If the C18:5 fatty acid was present, the authors should show if it is omega 3 that is not possible to identify by GC/MS. 

The total fatty acid content (Fig. 3) was not described in MM section.

There are no previous description of PKS, FAS, FA among others.

Inroduction – lines 74 -74 -please review

Figure 1 – It is not clear if the stirring was used in the ballons.

MM – lines 120 – 123- please re-write and inform: how many ballons of control and 13C isotope were used?”

MM – line 211 – please explain “C” in concentration µmol C L

Round 2

Reviewer 2 Report

The manuscript was improved and I think it is interesting for Biomolecules readers.